# DEEP LEARNING INFERENCES WITH HYBRID HOMOMORPHIC ENCRYPTION

## ABSTRACT

When deep learning is applied to sensitive data sets, many privacy-related implementation issues arise. These issues are especially evident in the healthcare, finance, law and government industries. Homomorphic encryption could allow a server to make inferences on inputs encrypted by a client, but to our best knowledge, there has been no complete implementation of common deep learning operations, for arbitrary model depths, using homomorphic encryption. This paper demonstrates a novel approach, efficiently implementing many deep learning functions with bootstrapped homomorphic encryption. As part of our implementation, we demonstrate Single and Multi-Layer Neural Networks, for the Wisconsin Breast Cancer dataset, as well as a Convolutional Neural Network for MNIST. Our results give promising directions for privacy-preserving representation learning, and the return of data control to users.

## 1 INTRODUCTION

The healthcare, finance, law and government industries often require complete privacy and confidentiality between various stakeholders and partners. With the advent of highly effective AI using deep learning, many real-world tasks can be made more effective and efficient in these industries. However deep learning approaches are seldom performed with privacy preservation in mind, let alone with the encryption of information throughout the entire process.

As a result, current deep learning implementations often cannot be used for these confidential applications. Homomorphic Encryption (HE) (Will & Ko, 2015) offers an opportunity to address the privacy preservation gap, for data processing in general and deep learning in particular. HE can be used to perform computation on encrypted information (Rivest et al., 1978), without ever having access to the plaintext information.

Our work combines the paradigms of deep learning and homomorphic encryption, allowing improved privacy for existing server-side models (Will et al., 2016), and thus enabling many novel, intelligent, privacy-guaranteeing services.

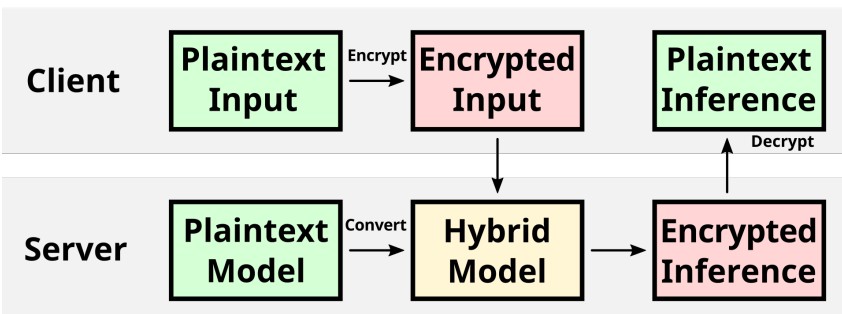

Figure 1: General overview of our privacy-preserving method for deep learning. Encrypted inputs are fed into our hybrid model on the server-side, and this produces encrypted outputs.

## 1.1 THE PROBLEM SCENARIO

Consider the following scenario: Some organization has created a deep learning solution, which can solve or assist in an important problem such as medical diagnosis, legal discovery or financial review. However they cannot release their model or parameters for client-side use, for fears that their solution could be reverse-engineered. At the same time, the client is unable or unwilling to send private information to other parties, which seemingly makes a server-sided application impossible.

To preserve the client's privacy, they would have to send their information encrypted, in such a way that only the client could decrypt it. Under a typical encryption scheme, the organization could not process this input in any meaningful way.

Any complete solution to this problem would require that the client's input is not revealed to the server, the server's model and weights are not revealed to the client, and that arbitrarily deep models with common operations, such as convolution and ReLU, can be supported by the server.

Our approach uses Fully Homomorphic Encryption (FHE), since the client's data can be processed confidentially, in any desired fashion. We will also learn that there are additional possibilities for making this approach efficient, without sacrificing security or functionality.

## 2 BACKGROUND

Both deep learning and FHE have seen significant advances in the past ten years, making them practical for use in production.

## 2.1 HOMOMORPHIC ENCRYPTION

Partially homomorphic encryption (PHE) has enabled multiplication (ElGamal, 1985) or addition (Paillier, 1999) between encrypted integers. However it was not possible to perform both of these operations under the same encryption system. This resulted in cryptosystems that were not Turing-complete. However the cryptosystem from Gentry et al. (2009) allowed both multiplication and addition to be performed flawlessly on encrypted integers. In theory, this system alone could be used to compute any arithmetic circuit. However in practice the bootstrapping operations were incredibly slow, taking up to 30 minutes in some cases (Gentry & Halevi, 2011).

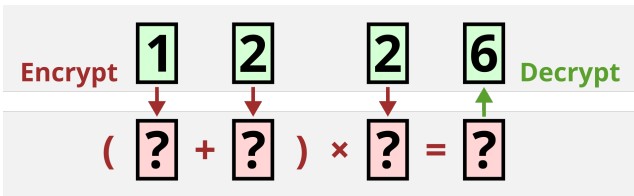

Figure 2: Simplified example of FHE.

## 2.2 FASTER HOMOMORPHIC ENCRYPTION

To improve the speed of FHE, several approaches were developed. The "bootstrapping" procedure is what allows for arbitrarily many sequential FHE operations. However the vast majority of time in any operation was spent performing bootstrapping, making it the key bottleneck.

Brakerski et al. (2014) made use of Leveled FHE, which eliminated the bootstrapping procedure, but limited the system to a finite number of sequential operations. Within these limits however, operations could be performed relatively quickly.

Brakerski et al. (2013) made use of Ciphertext Packing, allowing cryptosystems to pack many individual messages into a single encrypted message, increasing overall throughput.

FHEW (Ducas & Micciancio, 2015) takes a different approach, reducing the problem down to the NAND operation on binary numbers. This reduced the bootstrapping time to under 0.5 seconds.

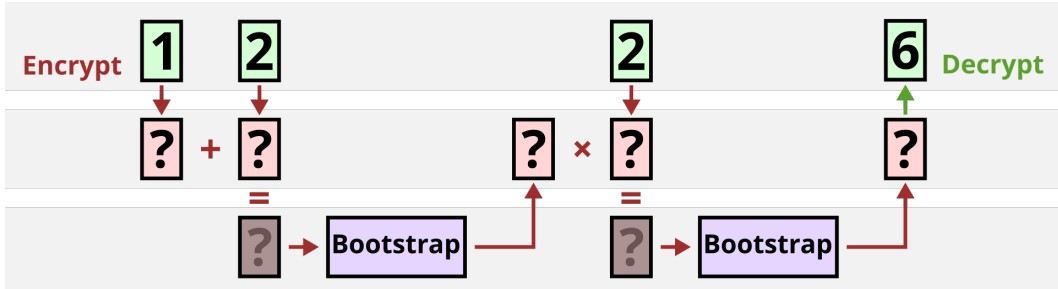

Figure 3: More detailed example of FHE. After performing FHE operations, ciphertexts may be bootstrapped, so that they can be used for subsequent operations. However this operation is costly.

TFHE (Chillotti et al., 2016) furthers this approach, reducing the bootstrapping time to under 0.1 seconds, while implementing many additional logic gates such as AND, OR, and XOR. Both FHEW and TFHE have open-source software implementations, and the FHEW implementation has since added some of the improvements and gates from Chillotti et al. (2016).

### 2.3 PRIVACY PRESERVING DEEP LEARNING

Mohassel & Zhang (2017) and Liu et al. (2017) do not use homomorphic encryption for online operations, and instead use a combination of garbled circuits and secret sharing. This approach requires more communication between the client and server(s), and can reveal the structure of a model to the client. Liu et al. (2017) proposes a form of model obfuscation to alleviate this, but no further details are given.

### 2.4 PRIVACY-PRESERVING MODEL TRAINING

Shokri & Shmatikov (2015) allows multiple parties to collaboratively build a model, by securely sharing training information. Mohassel & Zhang (2017) trains a model with secure weights, by obtaining secure training information from a client. We do not explore these concepts, since our problem scenario assumes that the server has exclusive access to the model structure and weights.

### 2.5 DEEP LEARNING WITH HOMOMORPHIC ENCRYPTION

Orlandi et al. (2007) implemented a number of old deep learning models under PHE, however this system computes activation functions in plaintext, compromising security and making it invalid for our problem scenario.

More recent work has implemented deep learning models under FHE (Xie et al., 2014; Dowlin et al., 2016). However these approaches have a variety of limits, such as a Leveled FHE system which limits model depth, and ineffective replacements for common activation functions. In general, these systems do not have the capacity to implement state-of-the-art models.

Dowlin et al. (2016) also uses plaintext parameters to reduce processing time, however the techniques used are specific to their cryptosystem. Our approach uses plaintext weights more directly, since we can exploit the properties of binary logic gates.

## 3 DESIGN

We decided to design deep learning functions for a binary FHE system, using bootstrapping. This is because deep learning operations can require arbitrarily many sequential operations, which would be difficult to implement with leveled FHE. Furthermore we want to support activation functions like ReLU, which can only be realistically achieved in FHE using a binary representation of the number. If we used a non-binary system, the number would have to be decomposed into binary for every comparison, which would be extremely inefficient.

Both TFHE and FHEW provide bootstrapped, boolean operations on binary inputs, with open-source software implementations, and thus we decided to support both of these, by abstracting shared concepts such as ciphertexts, logic gates, encryption, decryption and keys. We also modified FHEW to implement XOR, a critical logic gate for any efficient implementation of our design. This design should also be able to support any future binary FHE system.

## 3.1 HYBRID HOMOMORPHIC ENCRYPTION

We propose a general, hybridized approach for applying homomorphic encryption to deep learning (see Figure 1), which we call Hybrid Homomorphic Encryption (HHE). This approach greatly improves the efficiency and simplicity of our designs.

| A | B | A AND B | A | B | A NAND B | A | B | A XOR B |
|---|---|---------|---|---|----------|---|---|---------|
| 0 | 0 | 0 | 0 | 0 | 1 | 0 | 0 | 0 |
| 0 | 1 | 0 | 0 | 1 | 1 | 0 | 1 | 1 |
| 1 | 0 | 0 | 1 | 0 | 1 | 1 | 0 | 1 |
| 1 | 1 | 1 | 1 | 1 | 0 | 1 | 1 | 0 |

| A | B | A AND B | A | B | A NAND B | A | B | A XOR B |
|---|---|---------|---|---|----------|---|---|---------|
| 0 | ? | 0 | 0 | ? | 1 | 0 | ? | B |
| 0 | ? | 0 | 0 | ? | 1 | 0 | ? | B |
| 1 | ? | B | 1 | ? | NOT B | 1 | ? | NOT B |
| 1 | ? | B | 1 | ? | NOT B | 1 | ? | NOT B |

Figure 4: Output of various logic gates, along with equivalent gates where one input is unknown.

One important observation with binary bootstrapped FHE, is that a complete bootstrapping operation must be performed whenever all inputs to a logic gate are encrypted (ciphertexts). When the inputs to a logic gate are all unencrypted inputs (plaintexts), there is clearly no bootstrapping operation required. However when the input to a logic gate is one ciphertext and one plaintext, we still know enough information to not require a bootstrapping procedure.

Consider the NAND gate in Figure 4. If input A is a plaintext 1, then the output is always NOT B, while if the input is a plain 0, then the output is always 1. Even with the XOR gate in Figure 4, we can execute the gate without knowing any information about B, other than its inverse. Crucially, both FHEW and TFHE can perform the NOT operation almost instantaneously, without needing to perform the bootstrapping procedure.

Given our problem scenario, where the model is being hosted on a server, any parameters supplied by the server can be held in plaintext. Deep learning inferences largely involve multiplying inputs with static weights, so we can store the model weights in plaintext, and exploit this hybrid approach.

## 3.2 THE HYBRID FULL ADDER

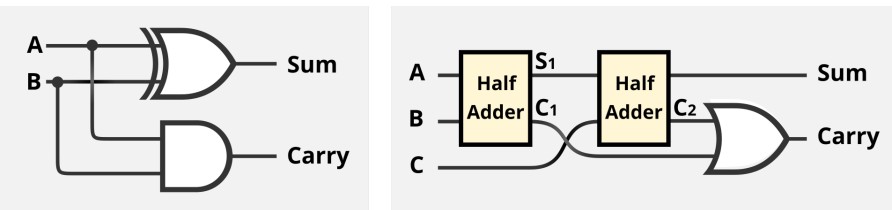

Figure 5: Typical half adder (left) and full adder (right).

In order to implement a deep learning model from logic gates, we must first build adder and multiplier circuits. Conveniently, these can largely be constructed out of a Hybrid Full Adder (HFA) circuit, as shown in Figure 6.

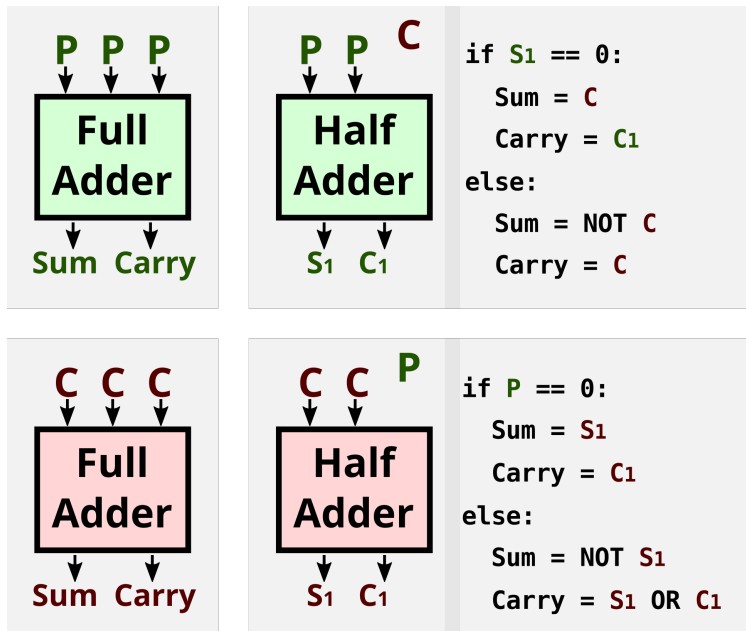

Figure 6: The four core components of the Hybrid Full Adder.

When three plaintexts or three ciphertexts are added, a typical plaintext or ciphertext full-adder can be used. However when two plaintexts and one ciphertext are added, we use a half-adder for the plaintexts, then approach the final sum and carry in a similar fashion to the hybrid XOR from Section 3.1. As a result, no bootstrapping homomorphic gates are needed, despite the use of one ciphertext.

When two ciphertexts and one plaintext are added, we can first half-add the ciphertexts together. If the plaintext is 0, we can just output the half-adder's sum and carry results. If the plaintext is 1, we can apply OR to the two carries. Here we only used two bootstrapping gates (from the half-adder), with one additional gate if the plaintext is 1.

These sub-units of the HFA cover all possible combinations of plaintexts and ciphertexts. In a number of cases, we use no bootstrapping gates, but in the worst case (all ciphertexts) we use 5 bootstrapping gates (from the full-adder).

### 3.3 Hybrid N-Bit Adder

An N-bit adder circuit is shown in Figure 7, which minimizes the number of gates required by using a half-adder for the first input, and using just two XORs for the final adder, when no overflow is required. Since we are creating a software implementation, we can iteratively loop through the middle section, enabling variable bit-depths.

If we make use of HFAs, the first carry-in can be a plaintext 0, while 2 Hybrid XORs can be used for the output. The implicit optimizations afforded by the HFA can be used to simplify a software implementation of this circuit, and we find that this also applies to other circuits with half-adders, full-adders or implicit plaintext bits.

In the realm of physical circuits, adder circuits can be made more efficient by parallelizing sections of the circuit. However this is not necessarily beneficial for our implementation, as the key source of improved performance will come from minimizing the number of bootstrapping gates.

### 3.4 Hybrid Baugh-Wooley Multiplier

We use the matrix from Baugh & Wooley (1973) to structure an efficient multiplier for two's complement numbers. First we produce partial products by applying AND to each bit in one input, against every bit in the other input. Figure 8 shows a 4-bit matrix where partial products, their complements

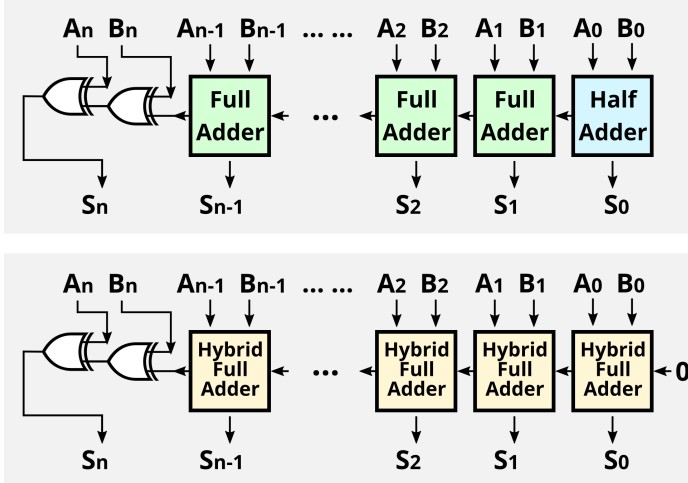

Figure 7: N-bit adder optimized to reduce the number of gates needed, and a hybrid variant.

and binary digits are included in specific locations. Each row is then added in turn, with the most significant bit of the final output being flipped. The result of this will be the multiplied number, with the combined bit-depth of the original numbers. Importantly, the matrix can be scaled to arbitrary bit-depths as explained in Baugh & Wooley (1973).

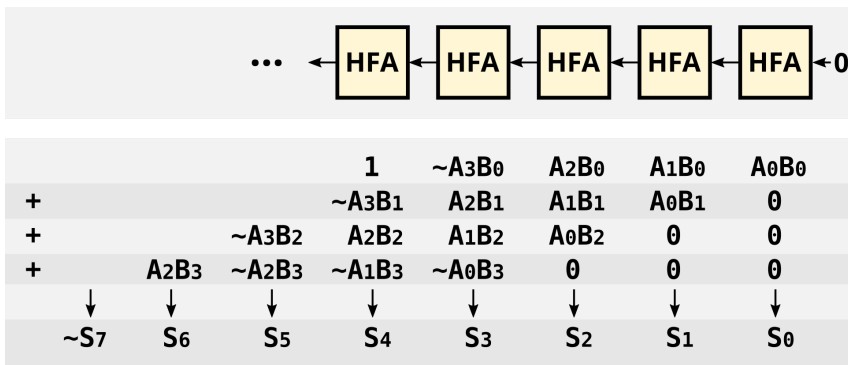

Figure 8: Baugh-Wooley matrix. A chain of Hybrid Full Adders can handle this efficiently.

Because the HFAs can efficiently handle plaintext inputs, we can populate the matrix with plaintext 0s and 1s. We want to include the overflow of each addition, so we can simply use an array of HFAs to add each row in turn, minimizing the number of bootstrapping operations, and leading to a surprisingly simple software implementation.

If we are multiplying one plaintext number with one ciphertext number, like in our problem scenario, it is important to consider the computation of partial products. Since 0 AND anything is 0, any time a 0 appears in the plaintext bit-string, that row in the matrix will be nearly all 0s, resulting in almost no bootstrapping operations. In fact when multiplying by 0 or $2^n$, almost no bootstrapping operations are required. Since more plaintext 0s in an input bit-string is clearly desirable, we two's-complement both the plaintext and the ciphertext, whenever the number of 1s in the plain bit-string is greater than the number of 0s. As a result, cases of $-2^n$ also make use of very few bootstrapping gates.

In general, the speed of this multiply operation is affected by not only the bit-depth of the input, but also the degree to which it is all-0s or all-1s. The worst-case plaintext input for this multiplier then, would be all numbers with bit-strings holding an equal number of 0s and 1s.

It is interesting to note that the properties of this hybrid Baugh-Wooley multiplier are similar to a plaintext Booth Multiplier (Booth, 1951), which excels on inputs with long "runs" of 1s or 0s. In

general, these properties could potentially be optimized for in deep learning models, where model weights are adjusted to increase the "purity" of the bit-strings.

Given the implicit optimizations afforded by the HFA inside both the adder and multiplier circuits, it is possible that other digital circuits could be implemented cleanly and efficiently with this approach.

### 3.5 ACTIVATION FUNCTIONS

Deep learning models typically apply the ReLU activation to intermediate layers, with a Sigmoid activation on the final layer. Variations such as the leaky ReLU or the hyperbolic tangent could also be used (Maas et al., 2013). Any differentiable, nonlinear function could be used as an activation function in principle, and Dowlin et al. (2016) uses the square $(x^2)$ activation in their FHE implementation, however this tends to result in less accurate models, or models which fail to converge. The square activation also requires that two ciphertexts be multiplied, which leads to a much slower operation.

---

**Algorithm 1** HybridReLU

Given a number $H$ made of HybridTexts, with sign bit $H_{sign}$:

1: **if** $H_{sign}$ is Plaintext **then**
2:     **if** $H_{sign} = 1$ **then**
3:         All other HybridTexts in $H \leftarrow$ plaintext 0
4: **else**                                                  ▷ $H_{sign}$ is a Ciphertext
5:     $S_{not} \leftarrow CipherNOT(H_{sign})$
6:     $HybridAND$ all other bits in H with $S_{not}$
7: $H_{sign} \leftarrow$ plaintext 0
8: **return** H

---

Because our numbers are represented in two's complement binary, ReLU can be implemented efficiently without approximation, as shown in Algorithm 1. A HybridText here refers to a bit being possibly plaintext or ciphertext, so a mixed bit-string is allowed.

As observed in Dowlin et al. (2016), if only the final layer uses the sigmoid activation function, we could potentially ignore this operation during model inference. Nevertheless, we can use piecewise approximation to represent it in our system.

Appendix A describes a fast approximation of the sigmoid activation, and more generally a method of using lookup tables to achieve piecewise approximation.

### 3.6 WEIGHTED SUMS AND CONVOLUTIONS

The main operation inside a simple neural network is the weighted sum, or the matrix multiply in the case of batch-processing. A simple weighted sum can be achieved by multiplying each input with a weight in turn, and adding it to an overall sum, using our adder and multiplier circuits. Implementing an efficient matrix multiplier could be valuable for future work.

The main operation inside a Convolutional Neural Network (CNN) is the convolution operation. We implement the naïve approach of performing a weighted sum of each pixel neighborhood, though more sophisticated implementations should be considered for future work, perhaps using fast Fourier transforms and matrix multiplies, as shown in Mathieu et al. (2013).

#### 3.6.1 MODEL OPTIMIZATION

Along with efficient implementations of digital circuits, we can also modify our deep learning models to require fewer bootstrapping operations. In general, we can use similar techniques to deep learning on embedded systems.

As discussed in Section 3.4, lower precision inputs and weights can be used to directly reduce the number of necessary operations. Because we support variable bit-depth, it could also be possible to optimize models such that more bits are allocated to weights that need them.

While deep learning frameworks generally use floating point values, the values used at any stage tend to be of similar scale, meaning we can instead use fixed-point arithmetic, and take advantage of our existing two's complement digital circuit designs.

Using separable convolutions can allow for very large kernels, at a comparable cost to traditional convolutions of small kernel size (Chollet, 2016). This simply requires convolving the horizontal component, followed by the vertical component, or vice-versa.

### 3.6.2 NORMALIZATION

While deep learning models can work well on low-precision inputs, most models require normalized inputs. If one were willing to give means and standard deviations to the client, this is a non-issue. However if normalization is to be performed server-side, many inputs can have means or standard deviations that cannot fit into low-precision numbers. The solution is to normalize these inputs at a higher bit-depth, such that the given inputs can be fully represented.

We can normalize encrypted inputs with our existing circuits, by adding the additive-inverse of the means, then multiplying by the multiplicative inverse of the standard deviations. The result can then be reduced to some desired precision for use in a model, ensuring that 3-7 standard deviations can still be represented.

## 4 IMPLEMENTATION

Our software is written in C++, and implements all the described functionality necessary to run the models in our results. We also implemented an interpreter, such that models constructed in the Keras framework (Chollet et al., 2015) can be exported and read in our software. Appendix B explains our software architecture in more detail.

### 4.1 DEEP LEARNING MODELS

To demonstrate this framework, we built a single-layer Neural Network (Perceptron), Multi-Layer Neural Network (MLP), and a CNN in Keras. We built and test our perceptron and MLP on the Wisconsin Breast Cancer dataset (Lichman, 2013), and the CNN for the MNIST database of handwritten digits (LeCun et al., 1998).

The CNN is designed to closely replicate the model in Dowlin et al. (2016), but we use ReLU instead of Square activations. They also apply multiple intermediate, unactivated layers, which are composed into a single intermediate layer for inference. However we avoided this step as it was unnecessary.

For the breast cancer dataset, 70% of instances (398) are used for training, while 30% of instances (171) are used for testing. Since the dataset contains more benign samples than malignant, we also weight malignant losses more than benign losses, to help balance the training process. For the handwritten digits dataset, 60,000 instances are used for training while 10,000 are used for testing.

Appendix C describes the structure and hyperparameters of each model in greater detail.

## 5 RESULTS

We measured the performance of various arithmetic and deep learning operations, then compared the speed and accuracy trade-offs for different model variants. All models and operations are executed on a single thread of an Ivy-Bridge i5-3570 CPU.

For comparison, we measure both the hybrid and ciphertext-only variants for the multiplier and adder circuits. However most other measured operations and models are hybrid, that is, we exploit efficiencies in plaintexts whenever possible.

We measure the average time for logic gate execution across 1,000 runs. Multiply and add operations are measured by the average from 16 runs, from initially chosen random numbers. The normalize operation time is an average for 1 input, from the 30 inputs for a Wisconsin breast cancer instance.

The execution times of the models are measured from single runs on a single instance. Since the normalization timings are shown separately, and since server-side normalization may not always be necessary (as discussed in section 4.1), the model execution times do not include the time spent normalizing inputs.

Some operations such as encryption, decryption and NOT were too fast to meaningfully measure, and had a negligible impact on model execution times, so these are not included in the results.

Appendix D provides additional execution timings, and expands upon these results.

## 5.1 PERFORMANCE

Table 1: Execution times for single numbers

| Operation | Bit-Depth | Time (FHEW) | Time (TFHE) |
|---|---|---|---|
| XOR gate | 1-bit | 66ms | **16ms** |
| Normalize | 16-bit | 22.2s | 5.0s |
| ReLU | 16-bit | 1.0s | **0.23s** |
| Cipher Multiply | 16-bit | 92.8s | 22.3s |
| Hybrid Multiply | 16-bit | 29.8s | **7.0s** |
| Cipher Multiply | 8-bit | 21.6s | 6.3s |
| Hybrid Multiply | 8-bit | 6.7s | **1.6s** |
| Cipher Multiply | 4-bit | 4.24s | 1.0s |
| Hybrid Multiply | 4-bit | 0.96s | **0.23s** |

Our measurements indicate that, at the time of this writing, the TFHE library performs operations around $4.1\times$ faster than the FHEW library. This gap could be wider on systems with FMA and AVX2 support, since TFHE has optional optimizations that utilize these instructions, however our Ivy-Bridge CPU does not support these.

From table 1, we show that for 16-bit inputs, our hybrid multiplier is around $3.2\times$ faster than an equivalent 16-bit ciphertext-only multiplier. However the 8-bit hybrid multiplier is a further $4.4\times$ faster than that, the 4-bit hybrid multiplier is $7.0\times$ faster than the 8-bit.

Interestingly, despite consisting of both a multiply and add operation, the average normalize operation is faster than an average multiply. This is because many of the standard deviations for the breast cancer inputs are actually very small, with only a few inputs requiring 16 bits or more. As a result, many standard deviations contain large numbers of plaintext 0s, which in turn leads to fewer homomorphic operations during the multiply. On the other hand, our multiply tests use random numbers, which should contain a more even mixture of 0s and 1s, and therefore execute more slowly.

For the breast cancer perceptron in table 2, we observe that the version with 8-bit intermediate values is almost $4.9\times$ faster than the 16-bit variant, while there is a $3.9\times$ speedup for the MLP. Importantly, these 8-bit speedups allow the CNN to execute in a somewhat feasible amount of time.

Table 1 also shows that the ReLU activation function is so fast, that its execution time becomes negligible relative to other operations. Unlike Dowlin et al. (2016), the vast majority of our performance bottleneck exists in the multiply and add operations of the weighted sums. This creates an interesting contrast, where our system can handle arbitrarily deep models with little overhead, but is

Table 2: Execution times for single model input instances

| Model | Bit-depth | Dataset | Time (FHEW) | Time (TFHE) |
|---|---|---|---|---|
| Perceptron | 16-bit | Cancer | 20m 42s | 4m 13s |
| Perceptron | 8-bit | Cancer | 3m 34s | 52s |
| MLP | 16-bit | Cancer | 2h 46m 20s | 37m 3s |
| MLP | 8-bit | Cancer | 43m 3s | 9m 34s |
| CNN | 8-bit | MNIST | - | 58h 11m 21s |

Table 3: Test accuracy and test loss for different models

| Model | Dataset | #Inputs | Bit-depth | #Parameters | Accuracy | Loss |
|---|---|---|---|---|---|---|
| Perceptron | Cancer | 30 | 16-bit | 31 | 97.7% | 0.075 |
| Perceptron | Cancer | 30 | 8-bit | 31 | 97.3% | 0.079 |
| MLP | Cancer | 30 | 16-bit | 257 | 98.2% | 0.067 |
| MLP | Cancer | 30 | 8-bit | 257 | 97.7% | 0.072 |
| CNN | MNIST | $28 \times 28$ | 8-bit | 93,708 | 99.0% | 0.048 |

slow when a large number of parameters are used, while Dowlin et al. (2016) only works on shallow models, but can handle a comparatively large number of parameters efficiently.

When examining model accuracies, there seems to be very little penalty for 8 bit model inputs and parameters, compared to 16-bit. However these accuracies are examined in Keras, which uses 32-bit floating point intermediate values. There could potentially be additional losses from our fixed-point arithmetic, which we might not have noticed, given that we only tested individual instances for correctness.

## 6 CONCLUSION

Our work shows that with the proposed Hybrid Homomorphic Encryption system, almost any production deep learning model can be converted, such that it can process encrypted inputs. Our design also makes it feasible to implement new or bespoke functionality, as the deep learning paradigm evolves.

Depending on the value of the problem and the size of the model, this system is already viable for production use. New and updated HE libraries appear frequently, and our code should adapt to any library which implements homomorphic logic gates. Therefore our software could potentially receive "free" performance gains, as the HE paradigm evolves.

### 6.1 FUTURE WORK

Having multi-threaded code would dramatically speed up our models, since individual weighted sum, convolution, multiply or add operations can be performed on separate threads. However even without multithreading, on a processor with $n$ cores, we could run $n$ duplicate models on $n$ input instances, and expect an $1/n$ amortized cost. Implementing an efficient matrix multiplier could also improve batch-processing times, and might allow for faster convolution operations.

Along with multithreading, there have also been efforts to design FHE systems around larger 3-bit circuits like full-adders (Biasse & Ruiz, 2015), and to accelerate existing HE libraries with GPUs (Lee et al., 2015). There is likely value in implementing homomorphic logic gates on GPUs, or even FPGAs, since almost every operation in our system can be made parallel in some form.

Plaintext weights are "encrypted" without noise in Dowlin et al. (2016) to improve performance. While both FHEW and TFHE are considerably different from this system, it may be possible to create a cryptosystem like FHEW that efficiently and directly integrates HHE.

In practice, the latest deep learning models use lots of additional functionality, such as residual blocks (He et al., 2016), or perhaps a self-normalizing activation function (Klambauer et al., 2017). It should be feasible to extend our implementation with most if not all of this functionality. Work has also been done to optimize deep learning models for embedded hardware, and we have not used all of the tricks available in this space – this would be another straightforward opportunity to build upon our approach.

Looking back at our problem scenario, an organization could now use our design, or even our software, and create a deep learning solution that performs powerful analysis, yet guarantees user privacy. Over time, this could potentially lead to an industry of privacy-centric software development, which can coexist with the expanding role of big data.

ACKNOWLEDGMENTS

Anonymous acknowledgements.

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

## A  APPENDIX: APPROXIMATING FUNCTIONS

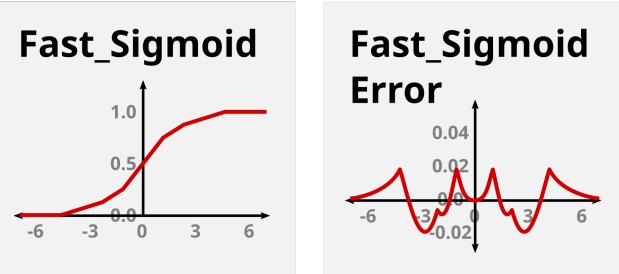

Figure 9: A piecewise linear approximation of Sigmoid, and its error.

---

**Algorithm 2** LookupTable

---

Given a number $C$ made of Ciphertexts, with sign-bit $C_{sign}$, and Components $Comp_1, \ldots, Comp_m$, with thresholds $Threshold_1, \ldots, Threshold_m$:

    **function** SIGNMASK($Cthresh, Cresult, Threshold$)
2:      $C_{upper} \leftarrow Cthresh_{sign}$
        $Cthresh \leftarrow Cthresh + Threshold$
4:      $C_{lower} \leftarrow NOT(Cthresh_{sign})$
        $C_{mask} \leftarrow C_{upper}\ AND\ C_{lower}$
6:      $Cresult \leftarrow$ AND each bit of $Cresult$ with $C_{mask}$
    $Cthresh \leftarrow C - Threshold_1$
8: $C_{lower} \leftarrow NOT(Cthresh_{sign})$
    $Comp_1 \leftarrow$ AND each bit of $Comp_1$ with $C_{lower}$
10: **for** $i \in \{2, \ldots, m-1\}$ **do**
        $Threshold_{current} \leftarrow Threshold_{i-1} - Threshold_i$
12:      SIGNMASK($Cthresh, Comp_i, Threshold_{current}$)
    $C_{upper} \leftarrow Cthresh_{sign}$
14: $Comp_m \leftarrow$ AND each bit of $Comp_m$ with $C_{upper}$
    **return** OR each bit of $Comp_1$ with matching bit from every other $Component$

---

**Algorithm 3** FastSigmoid

---

Given a number $C$ made up of Ciphertext bits:

    $Thresholds \leftarrow \{4, 2, 1, -1, -2, -4\}$
    $Component_1 \leftarrow 1$
3: $Component_2 \leftarrow (C \times 2^{-3}) + 2^{-1}$
    $Component_3 \leftarrow (C \times 2^{-2}) + 2^{-2}$
    $Component_4 \leftarrow (C \times 2^{-1})$
6: $Component_5 \leftarrow (C \times 2^{-2}) - 2^{-2}$
    $Component_6 \leftarrow (C \times 2^{-3}) - 2^{-1}$
    $Component_7 \leftarrow -1$
9: $Final \leftarrow$ LOOKUPTABLE($C, Components, Thresholds$)
    **return** $(Final + 1) \times 2^{-1}$

---

Lookup tables can be used to approximate functions. Depending on the function, one could use constants, linear equations, polynomials or even other functions as components for the lookup table, allowing for great versatility in our approach.

As shown in Figure 9, using a piecewise linear approximation of sigmoid with only factors of $2^n$, we can achieve an error of less than 0.02 against the original function. Algorithm 3 describes the sigmoid approximation, where a number of linear components are computed, and then applied to the lookup table according to a number of thresholds. Because we do not know what the input number

is, we must compute every component, so there is a trade-off between the number of components and the complexity of their calculation.

Algorithm 2 describes a lookup table, which selects one component, corresponding to a threshold for the input number. By subtracting the threshold from the number, and masking each component in turn with the resulting sign bits, we can isolate the correct component despite not knowing the plaintext value. At the end, all but one component will be entirely 0s, so each component can be ORed in turn to get the final result.

During our testing, we found that when training the perceptron for 100 epochs with the sigmoid output activation, then swapping in the "fast sigmoid" activation for 100 epochs, we achieve almost identical accuracies and losses, compared to just using sigmoid.

## B    APPENDIX: SOFTWARE ARCHITECTURE

We intend to release the implementation as open-source software, so in order to make it practical for others to use, we designed it to be easily configured and built. The architecture can largely be divided into four parts: the backend, arithmetic logic, deep learning and testing components.

The backend abstracts shared functionality between FHE libraries, and combines preprocessing macros with a CMake build script to switch between the desired backend and dependencies. Importantly, these features allow us to support any new backend with relative ease, so our software can take advantage of new or updated HE libraries over time. We also include a "Fake" backend which uses the same API, but does not actually perform encryption, for testing and debugging purposes.

The arithmetic logic implements our adder and multiplier circuits, which are supported by circuits such as the HFA, and assorted bit-manipulation functions.

The deep learning component implements the weighted sum, convolution, ReLU and normalization operations, as well as the interpreter, as described in the main text.

Finally, the testing component ensures that all digital circuits up to and including the HFA are executing exactly as intended, while the adder and multiplier circuits are tested on a fixed number of initially chosen random inputs. Deep learning operations and models are tested on single instances of their respective test datasets.

## C    APPENDIX: DEEP LEARNING MODELS

As described in the main text, we created three different models in the Keras framework. The structure of these models can be observed in Figure 10.

Each model is trained using the ADAM optimizer with Nesterov momentum (Kingma & Ba, 2014; Dozat, 2016), with a cross-entropy loss function and a batch-size of 128. The perceptron and MLP are trained for 200 epochs with a learning rate of 0.001, while the CNN is trained for 100 epochs with a learning rate of 0.002.

Since the perceptron has so few parameters to optimize, it may not always successfully converge on a first attempt. We repeatedly train the perceptron through the first 100 epochs, until it has found parameters that allow it to converge slightly. This may require anywhere from 1 to 5 attempts.

## D    APPENDIX: ADDITIONAL RESULTS

Continuing the trend from the main results, the 2-bit hybrid multiplier is $9.6\times$ faster than the 4-bit. Less dramatic speedups are observed for the add operations, where the 16-bit hybrid adder is $2.1\times$ faster than the cipher adder, and the 8-bit hybrid adder is $2.8\times$ faster than the 16-bit adder.

Table 4 also shows that compared to the hybrid multipliers, the cipher multipliers do not see as great a relative speedup. The 8-bit cipher multiplier is only $3.5\times$ faster than the 16-bit, and the 4-bit multiplier is only $6.3\times$ faster than the 8-bit. This can in part be explained by the fact that lower precision numbers are more likely to be of the form $2^n$ and $-2^n$, which our hybrid multiplier handles very quickly as discussed in section 3.4.

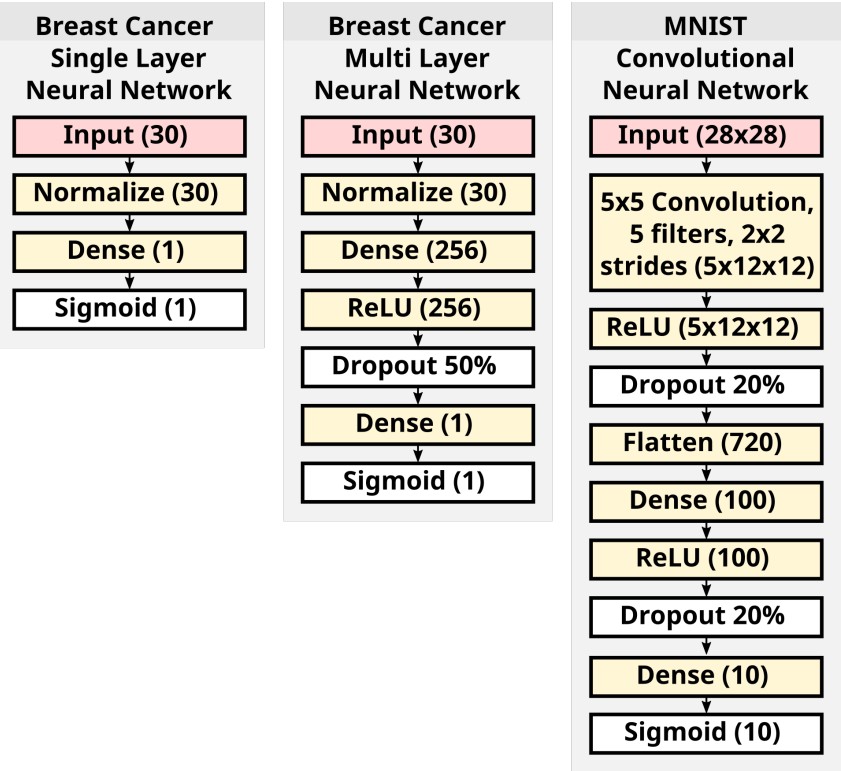

Figure 10: Structure of each implemented model. Brackets indicate the output shape for each layer. Sigmoid and Dropout operations are removed, when the models are exported to our framework.

Table 4: More Execution times for single numbers

| Operation | Bit-Depth | Time (FHEW) | Time (TFHE) |
|---|---|---|---|
| XOR gate | 1-bit | 66ms | 16ms |
| Normalize | 32-bit | 2m 1s | **28.7s** |
| Normalize | 16-bit | 22.2s | 5.0s |
| ReLU | 16-bit | 1.0s | 0.23s |
| Cipher Multiply | 32-bit | 7m 19s | 98.2s |
| Hybrid Multiply | 32-bit | 2m 18s | 34.9s |
| Cipher Multiply | 16-bit | 92.8s | 22.3s |
| Hybrid Multiply | 16-bit | 29.8s | 7.0s |
| Cipher Multiply | 8-bit | 21.6s | 6.3s |
| Hybrid Multiply | 8-bit | 6.7s | 1.6s |
| Cipher Multiply | 4-bit | 4.24s | 1.0s |
| Hybrid Multiply | 4-bit | 0.96s | 0.23s |
| Hybrid Multiply | 2-bit | 99ms | **24ms** |
| Cipher Add | 16-bit | 5.0s | 1.2s |
| Hybrid Add | 16-bit | 2.4s | 0.56s |
| Cipher Add | 8-bit | 2.3s | 0.53s |
| Hybrid Add | 8-bit | 0.82s | 0.20s |

