# OpenReview forum: "Deep Learning Inferences with Hybrid Homomorphic Encryption"
_ICLR.cc/2018/Conference — Reject_

### Official Review · AnonReviewer3 · 2017-11-26
**Deep Learning Inferences with Hybrid Homomorphic Encryption**

**Rating:** 4
**Confidence:** 5

**Review:**

Summary:
This paper proposes a framework for private deep learning model inference using FHE schemes that support fast bootstrapping.
The main idea of this paper is that in the two-party computation setting, in which the client's input is encrypted while the server's deep learning model is plain.
This "hybrid" argument enables to reduce the number of necessary bootstrapping, and thus can reduce the computation time.
This paper gives an implementation of adder and multiplier circuits and uses them to implement private model inference.

Comments:
1. I recommend the authors to tone down their claims. For example, the authors mentioned that "there has been no complete implementation of established deep learning approaches" in the abstract, however, the authors did not define what is "complete". Actually, the SecureML paper in S&P'17 should be able to privately evaluate any neural networks, although at the cost of multi-round information exchanges between the client and server.

Also, the claim that "we show efficient designs" is very thin to me since there are no experimental comparisons between the proposed method and existing works. Actually, the level FHE can be very efficient with a proper use of message packing technique such as [A] and [C]. For a relatively shallow model (as this paper has used), level FHE might be faster than the binary FHE.

2. I recommend the author to compare existing adder and multiplier circuits with your circuits to see in what perspective your design is better. I think the hybrid argument (i.e., when one input wire is plain) is a very common trick that used in the circuit design field, such as garbled circuit [B], to reduce the depth of the circuit.

3. I appreciate that optimizations such as low-precision and point-wise convolution are discussed in this paper. Such optimizations are very common in deep learning field while less known in the field of security.

[A]: Dowlin et al. Cryptonets: Applying neural networks to encrypted data with high throughput and accuracy.
[B]: V. Kolesnikov et al. Improved garbled circuit: free xor gates and applications.
[C]: Liu et al. Oblivious Neural Network Predictions via MiniONN transformations.

---

> ### Author Response · Authors · 2018-01-03
> **Reply to AnonReviewer3**
>
> Thank you for your constructive review!
>
> It is fair to challenge our claims that “there has been no complete implementation of established deep learning approaches”, because there have been some implementations of deep learning models whereby a server can perform inference, including SecureML, Cryptonets [A] and MiniONN [C]. With this in mind, it is important that we clarify our problem scenario. The server does not want to reveal the model to the client, and the client does not want to reveal the input to the server. While all of the given approaches secure the client input, only Cryptonets and our paper secure the model structure from a client, who may wish to reverse-engineer the model. MiniONN proposes obfuscating the model to alleviate this issue, but an implementation of this for arbitrary architectures is not given, would not be trivial, and would increase the number of client-server exchanges.
>
> Because SecureML and MiniONN are related works, we have updated the background section in our paper to discuss these works. We have also updated the problem scenario to more clearly explain what we consider to be a “complete implementation”.
>
> We agree that for a shallow model using message packing, leveled FHE could be faster than binary FHE, and conversely a leveled FHE would become impractical for sufficiently deep models.
>
> It is also important to note that Cryptonets uses the square activation instead of ReLU, and they present some disadvantages to this approach, in particular the unbounded derivative, making training difficult and limiting model depth. The square activation is also one of the most expensive operations in their network, because two ciphertexts must be multiplied.
> MiniONN can perform ReLU, but it does not use FHE, leading to other tradeoffs as discussed.
>
> We have updated the end of the results section, to better clarify the comparison between our work and Cryptonets, with regards to model size, depth and efficiency.
>
> We considered our circuits efficient in that they were much faster using a hybrid approach, compared to using only ciphertexts, and also that they allowed for an simpler implementation by abstracting plaintext, ciphertext and hybrid adders into a single unit. We have updated the “Hybrid Homomorphic Encryption” subsection of the design section, to better clarify that this is why we considered our approach efficient and simple.
>
> We have also updated the abstract, with the intention of toning down our claims, by using the language “no complete implementation of common deep learning operations, for arbitrary model depths, using homomorphic encryption”, and “efficiently implementing many deep learning functions with bootstrapped homomorphic encryption”. This should more cleanly cover the advantages of our work, compared to related literature, to the best of our knowledge.

---

### Official Review · AnonReviewer2 · 2017-11-26
**Review of "Deep Learning Inferences with Hybrid Homomorphic Encryption"**

**Rating:** 4
**Confidence:** 5

**Review:**

The paper presents a means of evaluating a neural network securely using homomorphic encryption. A neural network is already trained, and its weights are public. The network is to be evaluated over a private input, so that only the final outcome of the computation-and nothing but that-is finally learned.

The authors take a binary-circuit approach: they represent numbers via a fixed point binary representation, and construct circuits of secure adders and multipliers, based on homomorphic encryption as a building block for secure gates. This allows them to perform the vector products needed per layer; two's complement representation also allows for an "easy" implementation of the ReLU activation function, by "checking" (multiplying by) the complement of the sign bit. The fact that multiplication often involves public weights is used to speed up computations, wherever appropriate. A rudimentary  experimental evaluation with small networks is provided.

All of this is somewhat straightforward; a penalty is paid by representing numbers via fixed point arithmetic, which is used to deal with ReLU mostly. This is somewhat odd: it is not clear why, e.g., garbled circuits where not used for something like this, as it would have been considerably faster than FHE.

There is also a work in this area that the authors do not cite or contrast to, bringing the novelty into question; please see the following papers and references therein:

GILAD-BACHRACH, R., DOWLIN, N., LAINE, K., LAUTER, K., NAEHRIG, M., AND WERNSING, J. Cryptonets: Applying neural networks to encrypted data with high throughput and accuracy. In Proceedings of The 33rd International Conference on Machine Learning (2016), pp. 201–210.

SecureML: A System for Scalable Privacy-Preserving Machine Learning
Payman Mohassel and Yupeng Zhang

SHOKRI, R., AND SHMATIKOV, V. Privacy-preserving deep learning. In
Proceedings of the 22nd ACM SIGSAC Conference on Computer and Communications Security (2015), ACM, pp. 1310–1321.

The first paper is the most related, also using homomorphic encryption, and seems to cover a superset of the functionalities presented here (more activation functions, a more extensive analysis, and faster decryption times). The second paper uses arithmetic circuits rather than HE, but actually implements training an entire neural network securely.

 Minor details:

The problem scenario states that the model/weights is private, but later on it ceases to be so (weights are not encrypted).

"Both deep learning and FHE are relatively recent paradigms". Deep learning is certainly not recent, while Gentry's paper is now 7 years old.

"In theory, this system alone could be used to compute anything securely." This is informal and incorrect. Can it solve the halting problem?

"However in practice the operations were incredibly slow, taking up to 30 minutes in some cases." It is unclear what operations are referred to here.

---

> ### Author Response · Authors · 2018-01-03
> **Reply to AnonReviewer2**
>
> Thank you for your detailed review!
>
> We chose not to use a garbled circuit approach for our work, because this would reveal at least in part, the structure of the model. Part of our problem scenario is that the server does not wish to reveal the model to the client, and by extension the model’s structure.
>
> We do make comparisons between our work and Cryptonets, under the reference “Dowlin et al. (2016)”. It is fair to compare our paper with theirs, since they share a common goal. Their paper discusses three activation functions: sigmoid, ReLU and square. They do not attempt to implement sigmoid and ReLU, and instead use the square activation exclusively. Their paper presents the disadvantages to the square activation, in particular the unbounded derivative, making training difficult and limiting the depth of any model using this approach. It is also one of the most expensive operations in their network, because they must multiply two ciphertexts together. We have updated the “Activation Functions” subsection of the design section, to discuss the square activation in more detail.
>
> Because we use binary circuits, our approach can exactly replicate the ReLU activation, and a piecewise linear approximation of sigmoid. We implement both of these, however we did not feel that it was necessary to implement the square activation, because this was used in Cryptonets as a replacement for ReLU, to solve a problem unique to arithmetic circuits.
>
> We did not include decryption times in our results, because they were executing in less than a microsecond. Because our system only requires decryption at the very end of the process, it is a negligible cost compared to overall execution time. We have updated the results section to better clarify why we did not include these measurements.
>
> We did not feel that securely training a neural network, such as with SecureML, would be of benefit for our problem scenario. If a model is securely trained, all weights are restricted to those clients which provided training data, leading to a different scenario where the server hosts the model structure, the client provides the training data, and neither party has access to the weights. If the server chose to give the weights to the client, then the client could reconstruct the model and run it in plaintext, removing the need for the server.
> Similarly with “Privacy-preserving deep learning”, their goal is to have multiple parties collaboratively train a model, without revealing their respective training data. This is also leads to a different scenario, where each client has a local model.
> We have added a short “Privacy-Preserving Model Training” subsection to the background section, to reference these works and better clarify why we do not consider model training.
>
> It is fair to challenge the novelty of our work. As discussed, there have been a number of works which implement neural networks, and secure client inputs. However we feel that under our problem scenario, where the server does not wish to reveal the model to the client, and the client does not wish to reveal the input to the server, our approach is novel because it permits important functionality that is not present in Cryptonets, and allows the server to keep its model completely private, unlike SecureML and “Privacy-preserving deep learning”. We have updated the problem scenario, to hopefully prevent any ambiguity over what our goals were for this work.
>
> To address the minor details:
>
> By “weight privacy”, the intended message was that under our problem scenario, the server does not have to reveal the model structure or weights to the client. While they could encrypt their weights in our framework, it would substantially slow down operations as shown in our comparison between hybrid and ciphertext multipliers. We suggest that the weights are unencrypted, but are kept internal to the server. We have updated the problem scenario to more clearly state that the server does not reveal the model or weights to the client, as opposed to the server explicitly securing the weights.
>
> "Both deep learning and FHE are relatively recent paradigms". It is reasonable to consider deep learning and fully homomorphic encryption to be old paradigms. We have changed this sentence to “Both deep learning and FHE have seen significant advances in the past ten years”, reflecting that this work is built upon advances in the past decade.
>
> "In theory, this system alone could be used to compute anything securely." Indeed their system would not solve the halting problem! We have changed this to “compute any arithmetic circuit”.
>
> "However in practice the operations were incredibly slow, taking up to 30 minutes in some cases." We were referring to the time needed to run one bootstrapping operation, using an early implementation of Gentry’s FHE scheme. We have now clarified and referenced this.

---

### Official Review · AnonReviewer1 · 2017-11-27
**This paper proposes a hybrid Homomorphic  encryption system that is well suited for privacy-sensitive data inference applications with the deep learning paradigm. The research methodology is well organized, its rationale well explained and supports the stated problem resolution, the obtained results are interesting  and the paper is well written (commendable).**

**Rating:** 4
**Confidence:** 4

**Review:**

This paper proposes a hybrid Homomorphic encryption system that is well suited for privacy-sensitive data inference applications with the deep learning paradigm.
The paper presents a well laid research methodology that shows a good decomposition of the problem at hand and the approach foreseen to solve it. It is well reflected in the paper and most importantly the rationale for the implementation decisions taken is always clear.

The results obtained (as compared to FHEW) seem to indicate well thought off decisions taken to optimize the different gates' operations as clearly explained in the paper. For example, reducing bootstrapping operations by two-complementing both the plaintext and the ciphertext, whenever the number of 1s in the plain bit-string is greater than the number of 0s (3.4/Page 6).

Result interpretation is coherent with the approach and data used and shows a good understanding of the implications of the implementation  decisions made in the system and the data sets used.
Overall, fine work, well organized, decomposed, and its rationale clearly explained. The good results obtained support the design decisions made.
Our main concern is regarding thorough comparison to similar work and provision of comparative work assessment to support novelty claims.

Nota:
     - In Figure 4/Page 4: AND Table A(1)/B(0), shouldn't  A And B be 0?
     - Unlike Figure 3/Page 3, in Figure 2/page 2, shouldn't  operations' precedence prevail (No brackets), therefore 1+2*2=5?

---

> ### Author Response · Authors · 2018-01-03
> **Reply to AnonReviewer1**
>
> Thank you for your positive comments!
>
> To clarify, our work can extend both TFHE, FHEW, or any system implementing Fully Homomorphic Encryption over binary. As part of our results, we compare TFHE and FHEW, to help show that advances in this field will continue to benefit our work directly, since we can support any new system with minimal effort. We have updated the start of the design section in our paper, to make this statement more carefully.
>
> To address the notes:
>
> We have fixed the AND table in Figure 4, thank you for pointing this out.
>
> For Figure 2, we meant to show the operations getting applied in a linear order, but indeed 1+2*2=5. We have added brackets to Figure 2, and updated Figure 3 to show the process more clearly. Thank you again for finding this.

---

### Author Response · Authors · 2018-01-04
**Changes made to paper**

This comment provides a summary of all changes we have made to the paper.

We have fixed the AND table in Figure 4.

We have added brackets to Figure 2, and updated Figure 3 to show the Homomorphic Encryption process more clearly.

We have updated the “Activation Functions” subsection of the design section, to discuss the square activation in more detail.

We have updated the results section to better clarify why we did not include decryption timings.

We have updated the problem scenario to reduce ambiguity, to more clearly state that the server does not reveal the model or weights to the client (as opposed to the server explicitly securing the weights), and to more clearly explain what we consider to be a “complete implementation”.

We have added a short “Privacy-Preserving Model Training” subsection to the background section, to reference some related works, and better clarify why we do not consider model training.

We have added a short “Privacy-Preserving Deep Learning” subsection to the background section, to reference some works which do not use homomorphic encryption, and the trade-offs which result from this.

We have changed "Both deep learning and FHE are relatively recent paradigms" to “Both deep learning and FHE have seen significant advances in the past ten years”, reflecting that this work is built upon advances in the past decade.

For the sentence "In theory, this system alone could be used to compute anything securely." We have changed the end to “compute any arithmetic circuit”, better reflecting what Gentry's cryptosystem does.

For the sentence "However in practice the operations were incredibly slow, taking up to 30 minutes in some cases." We have clarified that this was for the bootstrapping operation in an implementation of Gentry's cryptosystem, and added a reference.

We have updated the end of the results section, to better clarify the comparison between our work and Cryptonets, with regards to model size, depth and efficiency.

We have updated the “Hybrid Homomorphic Encryption” subsection of the design section, to explain that this hybrid approach is why we consider our approach to be efficient and simple.

We have also updated the abstract, using the language “no complete implementation of common deep learning operations, for arbitrary model depths, using homomorphic encryption”, and “efficiently implementing many deep learning functions with bootstrapped homomorphic encryption”. This should more cleanly cover the advantages of our work, compared to related literature, to the best of our knowledge.

---

### Decision · Program_Chairs · 2018-01-29
**ICLR 2018 Conference Acceptance Decision**

**Decision:**

Reject

**Comment:**

While the reviewers all seem to think this is interesting and basically good work, the Reviewers are consistent and unanimous in rejecting the paper.
While the authors did provide a thorough rebuttal, the original paper did not meet the criteria and the reviewers have not changed their scores.